# Targeting resident astrocytes attenuates neuropathic pain after spinal cord injury

Qing Zhao[1,2†], Yanjing Zhu[1†], Yilong Ren[1,2†‡], Lijuan Zhao[1,2], Jingwei Zhao[1,2], Shuai Yin[1,2], Haofei Ni[1,2], Rongrong Zhu[1]*, Liming Cheng[1,2,3]*, Ning Xie[1,2]*

[1]Key Laboratory of Spine and Spinal Cord Injury Repair and Regeneration of Ministry of Education, Orthopaedic Department of Tongji Hospital, School of Medicine, School of Life Sciences and Technology, Tongji University, Shanghai, China; [2]Division of Spine, Department of Orthopedics, Tongji Hospital, Tongji University School of Medicine, Tongji University, Shanghai, China; [3]Clinical Center for Brain and Spinal Cord Research, Tongji University, Shanghai, China

*For correspondence:
rrzhu@tongji.edu.cn (RZ);
limingcheng@tongji.edu.cn (LC);
nxieprof18@tongji.edu.cn (NX)

†These authors contributed equally to this work

Present address: ‡Department of Orthopedics, Shanghai General Hospital, Shanghai Jiao Tong University School of Medicine, Shanghai, China

Competing interest: The authors declare that no competing interests exist.

## eLife Assessment

This **important** study demonstrated that ablation of astrocytes in the lumbar spinal cord not only reduced neuropathic pain but also caused microglia activation. The findings presented add considerable value to the current understanding of the role of astrocyte elimination in neuropathic pain, offering **convincing** evidence that supports existing hypotheses and insights into the interactions between astrocytes and microglial cells, likely through IFN-mediated mechanisms

**Abstract** Astrocytes derive from different lineages and play a critical role in neuropathic pain after spinal cord injury (SCI). Whether selectively eliminating these main origins of astrocytes in lumbar enlargement could attenuate SCI-induced neuropathic pain remains unclear. Through transgenic mice injected with an adeno-associated virus vector and diphtheria toxin, astrocytes in lumbar enlargement were lineage traced, targeted, and selectively eliminated. Pain-related behaviors were measured with an electronic von Frey apparatus and a cold/hot plate after SCI. RNA sequencing, bioinformatics analysis, molecular experiment, and immunohistochemistry were used to explore the potential mechanisms after astrocyte elimination. Lineage tracing revealed that the resident astrocytes but not ependymal cells were the main origins of astrocytes-induced neuropathic pain. SCI-induced mice to obtain significant pain symptoms and astrocyte activation in lumbar enlargement. Selective resident astrocyte elimination in lumbar enlargement could attenuate neuropathic pain and activate microglia. Interestingly, the type I interferons (IFNs) signal was significantly activated after astrocytes elimination, and the most activated Gene Ontology terms and pathways were associated with the type I IFNs signal which was mainly activated in microglia and further verified in vitro and in vivo. Furthermore, different concentrations of interferon and Stimulator of interferon genes (STING) agonist could activate the type I IFNs signal in microglia. These results elucidate that selectively eliminating resident astrocytes attenuated neuropathic pain associated with type I IFNs signal activation in microglia. Targeting type I IFNs signals is proven to be an effective strategy for neuropathic pain treatment after SCI.

## Introduction

SCI can cause sensorimotor deficits and autonomic changes, and approximately 65% of SCI patients suffer chronic, severe neuropathic pain (*Henwood and Ellis, 2004*). Long-term unrelieved pain will hinder the patient's rehabilitation training, delay functional recovery, and seriously affect the patient's

physical and mental health (*Henwood and Ellis, 2004*; *Masri and Keller, 2012*). Unfortunately, the effect of current treatments for neuropathic pain is limited (*Dworkin et al., 2003*). Exploring an effective treatment for neuropathic pain is a major clinical challenge. Astrocytes are the most abundant type of glial cell in the central nervous system (*Dozio and Sanchez, 2018*). Accumulating studies have shown that astrocytes play a key role in the induction and maintenance of persistent pain (*Ji et al., 2019*). Cytokines and chemokines produced by astrocytes can powerfully modulate excitatory and inhibitory synaptic transmission in pain circuits, leading to central sensitization, and a transition from acute to chronic neuropathic pain (*Ji et al., 2013*; *Ji et al., 2019*; *Kawasaki et al., 2008*). Inhibiting the proliferation of spinal astrocytes or glial fibrillary acidic protein (GFAP) expression could reduce neuropathic pain (*Ji et al., 2019*; *Tsuda et al., 2011*).

Astrocytes are mainly derived from resident astrocytes, NG2 progenitors, and ependymal cells (*Barnabé-Heider et al., 2010*; *Ren et al., 2017*; *Sofroniew, 2009*). SCI leads to massive proliferation of resident astrocytes, which are the most abundant and mainly located around the scar (*Barnabé-Heider et al., 2010*; *Sabelström et al., 2014*). Ependyma is regarded as the main source of endogenous neural stem cells, which widely migrate to injury areas and generate the majority of neuroprotective astrocytes after SCI (*Barnabé-Heider et al., 2010*; *Meletis et al., 2008*). However, only a small number of ependymal cells could proliferate and migrate to the lesion score after SCI (*Ren et al., 2017*). NG2 cells, also known as oligodendrocyte progenitor cells, were previously considered to be the component of glial scar after SCI (*Church et al., 2016*; *Hackett and Lee, 2016*; *Hesp et al., 2015*). While, reports about their functions are still controversial, partly because there are several cell types that up-regulate proteoglycan NG2 after SCI, such as oligodendrocyte progenitor cells, pericytes, and macrophages (*Huang et al., 2020*; *Nishiyama et al., 2005*). Until now, there is limited data on the main origins of astrocytes-induced neuropathic pain after SCI and whether selectively targeting these astrocytes could attenuate neuropathic pain is still unclear.

Cervical and thoracic SCI often induce below-level neuropathic pain (*Gwak et al., 2012*; *Shiao and Lee-Kubli, 2018*; *Vierck, 2020*), partly through activating signal transducer and activator of transcription 3 (STAT3)-dependent reactive astrocytes (*Ono et al., 2020*). The expression of tumor necrosis factor-α (TNF-α), interleukin-1 β (IL-1β), and interleukin-6 (IL-6) in lumbar enlargement was closely associated with below-level neuropathic pain (*Honjoh et al., 2019*). Mainly because the ascending tracts and superficial dorsal horn (SDH) in lumbar enlargement are closely associated with proprioception and pain signal processing of the lower limbs (*Watson, 2012*). Thus, targeting the astrocytes in lumbar enlargement may improve neuropathic pain after cervical or thoracic SCI.

A C57BL/6 mouse neuropathic pain model with spinal cord contusion at thoracic level 10 (T10) was constructed. Pain behaviors were measured by an electronic von Frey apparatus and a cold/hot plate. Transgenic mice and lineage tracing were used to determine the main origins of astrocytes-induced neuropathic pain after SCI. To investigate if eliminating these astrocytes in lumbar enlargement could attenuate neuropathic pain after SCI, the transgenic mice, adeno-associated virus (AAV) vector, and diphtheria toxin (DT) were used. A previously undescribed improvement effect of targeting these astrocytes on neuropathic pain was identified. Combining with RNA sequencing, bioinformatics analysis, immunohistochemistry, molecular experiment, and cultured microglial cells, the potential mechanisms of pain relief were further explored after selective astrocytes elimination.

## Results

### Resident astrocytes but not ependymal cells were the main origins of astrocytes-induced neuropathic pain after SCI

The construction strategies of transgenic mice are described in *Figure 1A*. The brain slices from C57BL/6-*Foxj1*[em1(GFP-CreERT2-polyA)Smoc] mice showed that choroids plexus in both the fourth ventricle and lateral ventricle had been labeled by reporter gene EGFP (*Figure 1B*). However, EGFP-positive cells were not found in the central canal of the spinal cord under normal conditions (data not shown), indicating that ependymal cells may not be activated. Therefore, C57BL/6-*Foxj1*[em1(GFP-CreERT2-polyA)Smoc] mice were used to establish contusion SCI models and examine the activation and differentiation of ependymal cells. As reported before (*Ren et al., 2017*), ependymal cells did not transform into astrocytes in the lesion area (*Figure 1C*). S100 family protein p11 (S100A10) is previously regarded as a marker of neuroprotective astrocytes (*Liddelow et al., 2017*), but not unique and is expressed by

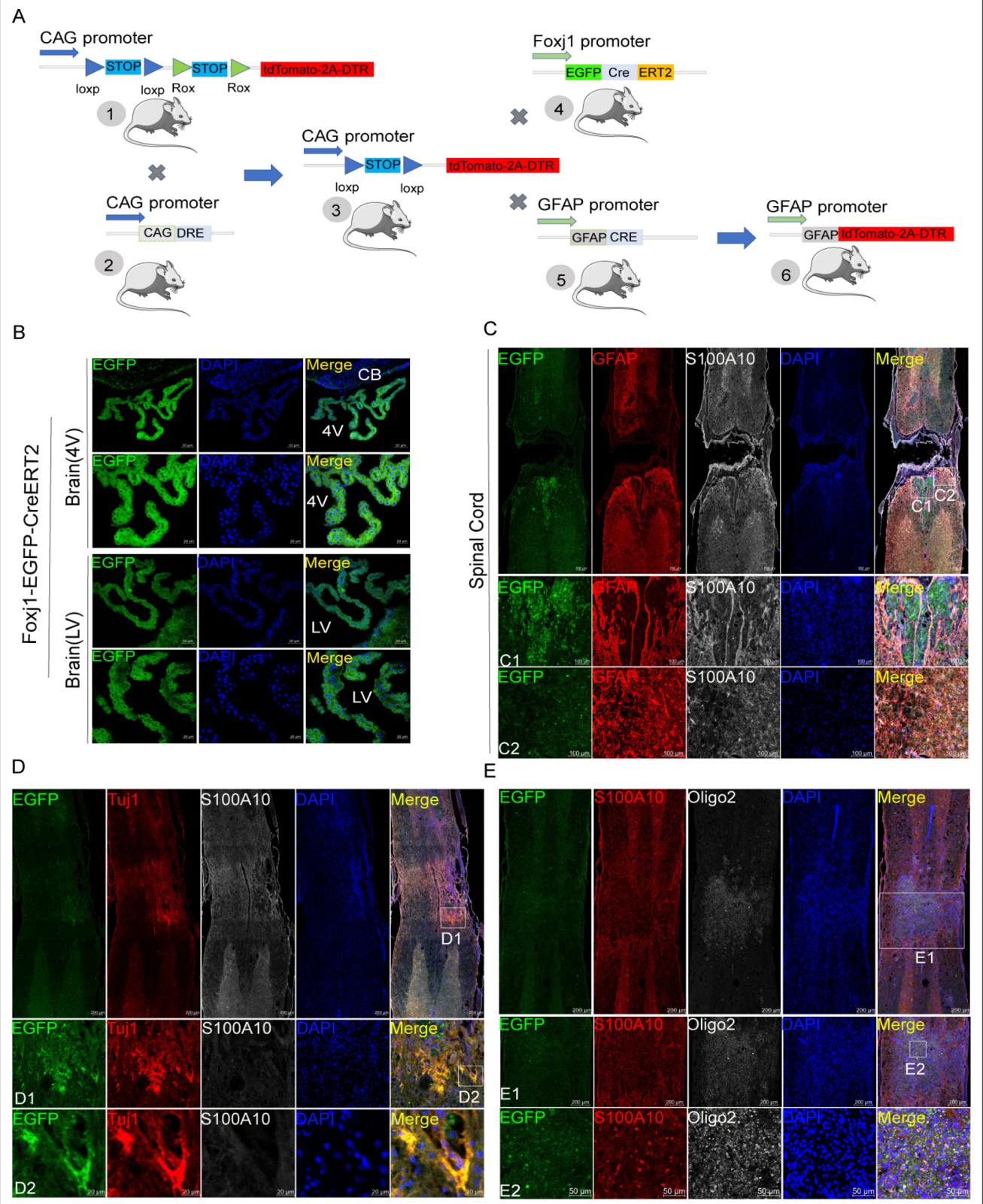

**Figure 1.** Ependymal cells did not transform into astrocytes in our transgenic mice. (**A**) Construction strategies of transgenic mice. (**B**) Ependymal cells both in the fourth ventricle (4 V) and lateral ventricle (LV) had been labeled by reporter gene eGFP (green). n=6 biological repeats. (**C**) Ependymal cells did not produce astrocytes in the lesion area. n=3 biological repeats. (**D–E**) Ependymal cells were observed to produce offspring of neurons (Tuj1+,

*Figure 1 continued on next page*

Figure 1 continued

red), Oligo2 (white) negative, and S100A10 (white in **D** or red in **E**) positive cells in the epicenter. n=3 biological repeats. 4V=fourth ventricle, LV = lateral ventricle, CB = Cerebellum. Scale bars had been indicated in pictures.

The online version of this article includes the following figure supplement(s) for figure 1:

**Figure supplement 1.** S100 family protein p11 (S100A10) is expressed by various cell types.

various cell types (**Figure 1—figure supplement 1A–C**). Instead, ependymal cells were observed to produce offspring of Tuj1 positive neurons, Oligo2 negative and S100A10 positive cells in epicenter (**Figure 1D–E**). Totally, ependymal cells did not transform into astrocytes as well as Oligo2 positive oligodendrocytes. Therefore, the resident astrocytes but not ependymal cells were the main origins of astrocytes-induced neuropathic pain after SCI.

## Using transgenic mice and AAV vectors could selectively target resident astrocytes in lumbar enlargement

The experiment flowchart is described in **Figure 2A**. Compared with the sham group, the SCI group showed increased astrocytes activation (GFAP expression) in ventral spinothalamic tracts (VST, 65.08±2.25 vs 52.71±1.02), SDH (59.62±2.21 vs 49.05±1.30, **Figure 2B** and **Figure 2—figure supplement 1A**), dorsal spinocerebellar tract (DSC, 81.64±3.63 vs 51.69±1.22), gracile fasciculi (GR, 58.37±1.89 vs 49.63±0.83) and ventral spinocerebellar tract (VSC, 75.73±2.21 vs 50.88±1.29). Thus, astrocyte activation in the lumbar enlargement is associated with neuropathic pain after SCI. AAV2/5-GfaABC1D-Cre with a minimal GFAP promoter was used to selectively target resident astrocytes of $Gt(ROSA)26Sor^{em1(CAG-LSL\ -tdTomato-2A-DTR)Smoc}$ mice. AAV2/5-GfaABC1D-Cre (1:3 dilution, ≥0.33E+13 V.G/ml) was injected into the spinal cord of mice (n=3, **Figure 2—figure supplement 1B**). Two weeks after the AAV2/5-GfaABC1D-Cre injection, immunofluorescence staining showed that a large number of astrocytes were labelled (**Figure 2—figure supplement 1B**). Meanwhile, almost no tdTomato-positive cells were found in NEUN-positive neurons or Oligo2-positive oligodendrocytes (**Figure 2—figure supplement 1B** and unshown data). To explore the optimal virus concentration, different dilutions including 1:10 (≥1E+12 V.G/ml), 1:100 (≥1E+11 V.G/ml), 1:250 (≥0.4E+11 V.G/ml), 1:500 (≥0.2E+11 V.G/ml), 1:1000 (≥1E+10 V.G/ml) and 1:10000 (≥1E+9 V.G/ml), were set for injection. Immunofluorescence results of spinal cord tissue and slices revealed that the dilution of 1:100 (≥1E+11 V.G/ml) was the optimal intervention concentration for concentrations below 1:100 can hardly target astrocytes (**Figure 2—figure supplement 1B** and unshown data). Then AAV2/5-GfaABC1D-Cre (≥1E+11 V.G/ml) was injected into $Gt(ROSA)26Sor^{em1(CAG-LSL\ -tdTomato-2A-DTR)Smoc}$ mice 2 weeks before T10 contusion. The labeled resident astrocytes were mainly located on the dorsal side of the lumbar enlargement (**Figure 1C**). It was also found that the labeled astrocytes by AAV2/5-GfaABC1D-Cre accounted for 39.03±1.97% astrocyte on the dorsal side of SDH (**Figure 1D**). All labeled astrocytes in the SCI + AAV + DT group were eliminated after DT injection (**Figure 1C**) and the expression of GFAP was significantly decreased on the dorsal side after resident astrocytes elimination (**Figure 1D**).

## Selective resident astrocytes elimination in lumbar enlargement attenuated neuropathic pain

Pain-related behaviors were measured with an electronic von Frey apparatus and a cold/hot plate, including mechanical allodynia, and thermal and cold hyperalgesia. The pain test was carried out as described in the experiment flowchart (**Figure 2A**). After contusion, the mice showed a complete hindlimb paralysis and weight loss, and these effects gradually recovered (**Figure 3A–B** and **Figure 3—figure supplement 1**). The 5 g impactor and 6.25 mm height contusion could induce mice to obtain obvious pain symptoms (**Figure 3C–F**). Mechanical allodynia (reflected by paw withdrawal threshold) occurred at 14 dpi (day post-injury, left hindlimb cut-offs vs. baseline; right hindlimb cut-offs vs. baseline), peaked at 28 dpi and sustained until 43 dpi (**Figure 3C–D** and **Figure 3—figure supplement 2**). Cold hyperalgesia (reflected by paw withdrawal frequency) and thermal hyperalgesia (reflected by paw withdrawal latency) also occurred at 14 dpi, and peaked at 43 dpi (**Figure 3E–F** and **Figure 3—figure supplement 2**). There was no statistical significance in pain-related behaviors, weight and motor function between the sham group and the sham AAV group (**Figure 3A–F** and **Figure 3—figure supplement 3**).

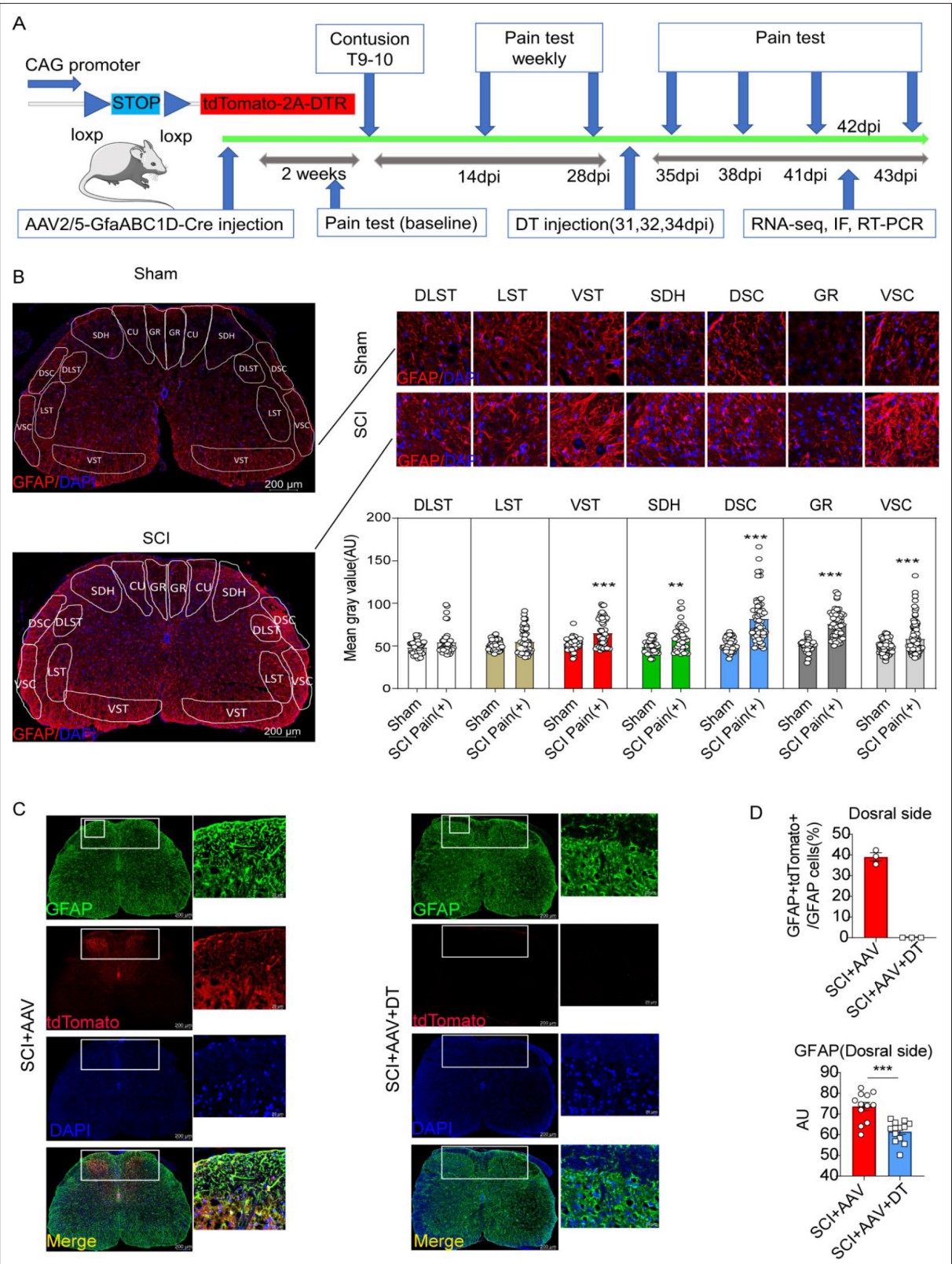

**Figure 2.** Resident astrocytes in lumbar enlargement were selectively targeted and eliminated. (**A**) The experiment flowchart. (**B**) The histogram and statistical results of glial fibrillary acidic protein (GFAP) fluorescence intensity. The location of the three largest ascending tracts of lumbar enlargements, including (1) CU, GR; (2) LST, DLST, VST; (3) DSC, VSC, along with SDH and gray matter. CU = cuneate fasciculus, GR = gracile fasciculus, LST = lateral spinothalamic tract, DLST = dorsolateral spinothalamic tract, VST = ventral spinothalamic tract, DSC = dorsal spinocerebellar tract, VSC = ventral

*Figure 2 continued on next page*

*Figure 2 continued*

spinocerebellar tract, SDH = superficial dorsal horn. n=3 biological repeats. Scale bar, 200 μm. Values are the mean ± SEM. Statistical significance was determined by two-way ANOVA followed by the Student Newman–Keuls post hoc test. **p<0.01, ***p<0.001, Sham group vs. spinal cord injury (SCI) Pain positive (+) group. (**C**), AAV2/5-GfaABC1D-Cre targeted astrocytes were mainly on the dorsal side and reflected by tdTomato and GFAP double-positive cells in the SCI + AAV group, no tdTomato, and GFAP double-positive cells were found in the SCI + AAV + DT group. n=3 biological repeats. Scale bars had been indicated in pictures. (**D**), the efficiency of astrocyte labeling and elimination in the dorsal side of spinal cord. Immunofluorescence analysis results of the GFAP after selective astrocyte elimination. Values are the mean ± SEM. Statistical significance was determined by two-way ANOVA followed by the Student Newman–Keuls post hoc test. n=3 biological repeats. TdTomato-positive cells indicated AAV. 2/5-GfaABC1D-Cre targeted astrocytes. ***p<0.001.

The online version of this article includes the following figure supplement(s) for figure 2:

**Figure supplement 1.** Astrocytes activation in the lumbar enlargement of neuropathic pain mice.

A total of 36 mice with neuropathic pain were randomly selected to receive diphtheria toxin (DT) injection on 31, 32, and 34 dpi (male = 18, female = 18, SCI + AAV + DT group), and the 36 mice left were classified as a control group (male = 18, female = 18, SCI + AAV group). Then pain tests were carried out on 35 dpi after DT injection. Motor function (BMS score) and weight did not change significantly in mice of the SCI + AAV + DT group after selective resident astrocytes elimination in the lumbar enlargement (*Figure 3A–B* and *Figure 3—figure supplement 1*). Mechanical allodynia of the left hindlimb attenuated after selective astrocytes elimination at 35 dpi (first day after astrocytes elimination, *Figure 3C* and *Figure 3—figure supplement 2*). The pain attenuation effects peaked at 41 dpi (seventh day after astrocytes elimination, *Figure 3C*). The mechanical allodynia of the right hindlimb was relieved after selective astrocytes elimination at 38 dpi (fourth day after astrocytes elimination), and the pain attenuation effects peaked at 41 dpi (*Figure 3D* and *Figure 3—figure supplement 2*). The attenuation of both cold and thermal pain occurred at 38 dpi, and the pain attenuation effects peaked at 41 dpi (*Figure 3E–F* and *Figure 3—figure supplement 2*). To further determine the effect of selective astrocytes elimination on hindlimb pain, the pain parameters on left and right hindlimbs were measured, and observed no obvious difference between them (data not shown). Furthermore, the effect of sex on neuropathic pain, and observed with no significant differences between male and female mice (*Figure 3—figure supplements 4–6*). In addition, the effect of DT injection alone on neuropathic pain of wild-type mice was examined after contusion. The contusion models were constructed as mentioned above. The pain response was similar in wild-type mice with or without DT injection (*Figure 3—figure supplement 7*).

## Microglia were activated after selective resident astrocytes elimination in lumbar enlargement

Immunofluorescence staining showed that increased activation of astrocytes and microglia and neuron loss were observed in the lesion area in the SCI + AAV group and SCI + AAV + DT group after contusion (*Figure 4A and C* and *Figure 4—figure supplement 1*). Compared with the SCI + AAV group, enhanced expression of microglia marker Ionized calcium-binding adaptor molecule 1 (IBA1) in lumbar enlargement was observed after selective resident astrocytes elimination in the SCI + AAV + DT group (*Figure 4B–C*). Furthermore, compared with the SCI + AAV group, the heatmap of RNA-seq analysis results showed that the transcripts of pro-inflammatory microglial marker genes elevated in the SCI + AAV + DT group (*Figure 4D*). As previously reported (*Watson, 2012*), the three largest ascending tracts, gray matter, and SDH were selected to analyze in the immunohistochemistry images. Selective resident astrocytes elimination led to enhanced IBA1 expression in the left DSC, left deep gray matter, left LST, left SDH, left VSC, right GR, right shallow gray matter, and right SDH (*Figure 4E–H*). These results indicated that selective resident astrocytes elimination activated the microglia in lumbar enlargement.

## Selective resident astrocytes elimination activated the type I IFNs signal in lumbar enlargement

To explore the potential mechanisms of pain attenuation after selective resident astrocytes elimination, RNA-seq and bioinformatics analysis were conducted. RNA-seq analysis revealed that there were few DEGs between the sham group and sham AAV group (*Figure 5—figure supplement 1A*). Between the SCI + AAV group and the sham AAV group, 1872 differentially expressed genes (DEGs) were

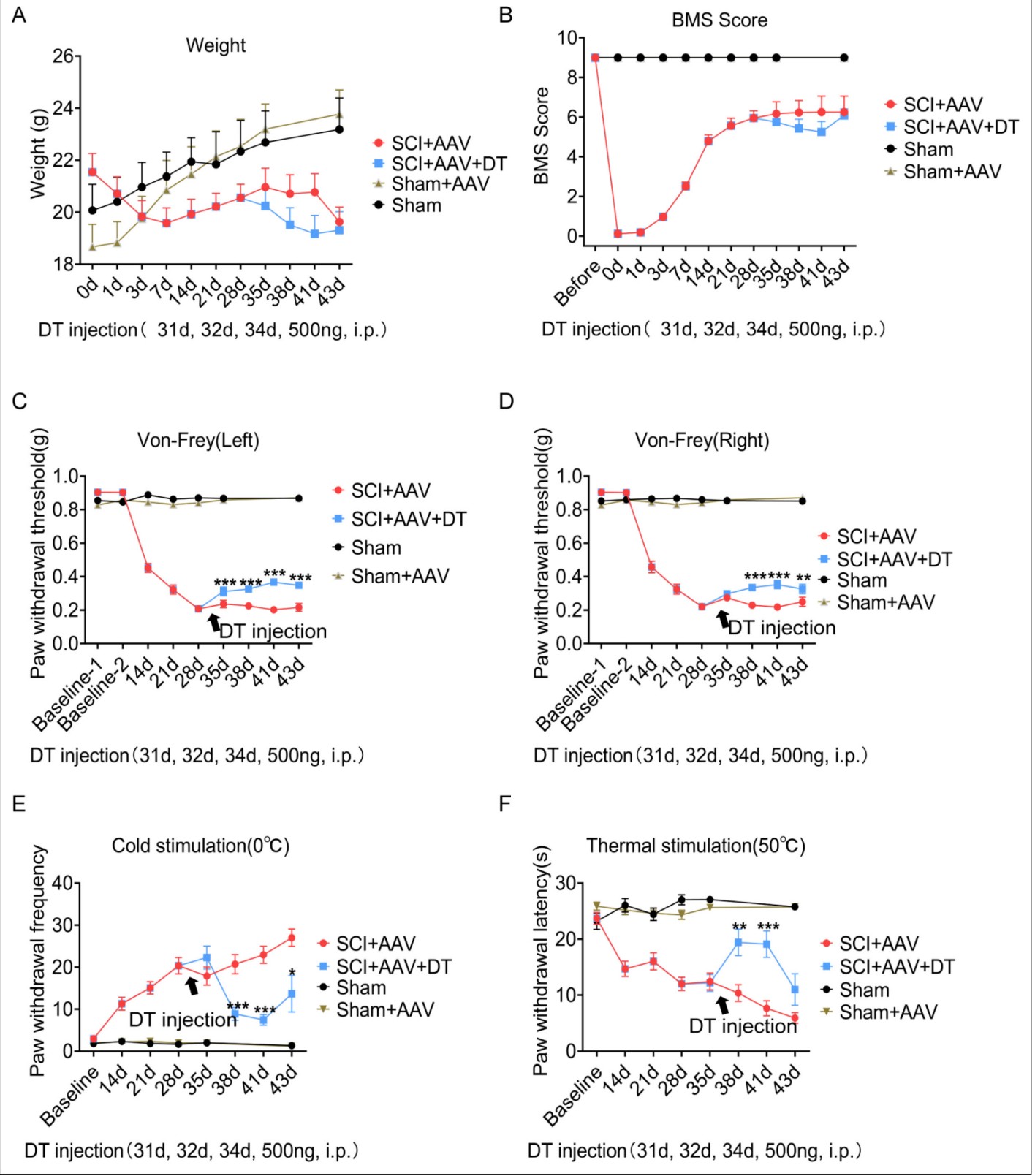

**Figure 3.** Selective astrocyte elimination in lumbar enlargement attenuated neuropathic pain. Change of weight (**A**) BMS scores (**B**), mechanical allodynia (**C–D**), cold hyperalgesia (**E**), and thermal hyperalgesia (**F**) in mice after astrocyte elimination. Sham group, n=30; sham + AAV group, n=30; SCI + AAV group, n=36; SCI + AAV + DT group, n=36. Arrow, 500 ng DT injection was performed on 31, 32, and 34 dpi (days post-injury). Left, left hindlimbs;

*Figure 3 continued on next page*

*Figure 3 continued*

right, right hindlimbs. i.p., intraperitoneal injection. Values are the mean ± SEM. Statistical significance was determined by two-way ANOVA followed by the Student Newman–Keuls post hoc test. *p<0.05, **p<0.01, ***p<0.001, SCI + AAV + DT group vs. SCI + AAV group.

The online version of this article includes the following figure supplement(s) for figure 3:

**Figure supplement 1.** Selective astrocyte elimination in lumbar enlargement attenuated neuropathic pain.

**Figure supplement 2.** Selective astrocyte elimination in lumbar enlargement attenuated neuropathic pain.

**Figure supplement 3.** Selective astrocyte elimination in lumbar enlargement attenuated neuropathic pain.

**Figure supplement 4.** No differences in neuropathic pain were observed between male and female mice.

**Figure supplement 5.** No significant differences in neuropathic pain were observed between male and female mice.

**Figure supplement 6.** No significant differences in neuropathic pain were observed between male and female mice.

**Figure supplement 7.** Pain symptoms were similar in wild-type mice with or without Diphtheria toxin (DT) injection.

found in the SCI + AAV group, including 1654 upregulated and 218 downregulated genes, as shown in the volcano map (*Figure 5—figure supplement 1B*). The Gene ontology (GO) enrichment analysis and KEGG pathway analysis revealed that the abundance of DEGs enriched in several biological processes, which were associated with immune responses including immune system processes, inflammatory responses, and innate immune responses (*Figure 5—figure supplement 1C–D*). Compared with the SCI + AAV group, the SCI + AAV + DT group (astrocytes elimination group) had 1120 DEGs, including 688 upregulated and 432 downregulated genes. The upregulated DEGs after selective resident astrocytes elimination were mainly enriched in regulating interferons (IFNs) production and signal transduction of type I IFNs (*Figure 5A*; *Borden et al., 2007*; *Owens et al., 2014*; *Rothhammer et al., 2016*). Protein-protein interaction (PPI) network analysis showed the interaction of these DEGs (*Figure 5B*). Variability was observed between individual animals within the SCI + AAV and SCI + AAV + DT groups, but the expression differences of DEGs in these two groups were significant.

GO enrichment analysis showed that the dominant biological processes were associated with response to virus, immune and type I IFNs responses, including defense response to virus, response to virus, immune system process, cellular response to interferon-alpha/-beta, positive regulation of interferon-alpha/-beta production or secretion, type I interferon signaling pathway, response to interferon-beta, type I interferon biosynthetic process (*Figure 5C*). KEGG pathway analysis found that several signaling pathways were disturbed, including the influenza A, Epstein-Barr virus infection, Herpes simplex virus 1 infection, Measles, Hepatitis C, and NOD-like receptor signaling pathways (*Figure 5D*). Heatmap showed significantly elevated transcripts level of critical genes involved in the type I IFNs signaling pathway, including *Jak1/2/3, Ifna16, Cgas, Tyk2, Ifnar1/2, Sting1, Irf3/7/9, Ifnb1*, and *Ifih1* (*Figure 5E*). Meanwhile, gene set enrichment analysis (GSEA) showed the disturbance of the Janus kinase (Jak)- STAT signaling pathway and cytokine–cytokine receptor interactions (*Figure 5F–G*). Thus, GO enrichment, KEGG pathway, and GSEA analysis results were associated with the type I IFNs signaling pathway (*Barrat et al., 2019*; *Ivashkiv and Donlin, 2014*). Totally, these results indicated that selective resident astrocytes elimination activated the type I IFNs signal in lumbar enlargement.

## The type I IFNs signal was activated in microglia and associated with pain relief

In order to further verify the results of RNA-seq, RT-PCR was carried out and the results showed the mRNA expression of genes associated with the type I IFNs signaling pathway increased, including *Ifih1(Mad5), Irf3, Irf7, If144, Stat1, Stat2, Parp14, Irf9, Isg15, Oas2, Oas3, Zbp1, Ifit1, Usp18, Dhx58*, and *Eifzak2* (*Figure 6A*). The mRNA expression of type I IFNs signaling genes was similar in wild-type mice with or without DT injection (*Figure 6B*). Furthermore, three critical proteins involved in the type I IFNs signal with significantly increased expression at the transcriptional level were selected, including IRF7, ISG15, and STAT1. Immunofluorescence analysis found that the expression of IRF7, ISG15, and STAT1 proteins was also significantly increased after selective astrocytes elimination (*Figure 6C–D*).

Although various cells in the central nervous system can synthesize and secrete interferons, microglia are resident immune cells and are believed to be the main source of IFN-β within the central nervous system during inflammatory conditions (*Owens et al., 2014*). Reinert and colleagues (*Reinert et al., 2016*) identified microglia as the primary producers of type I IFNs. ISG15 and IRF7 were co-located

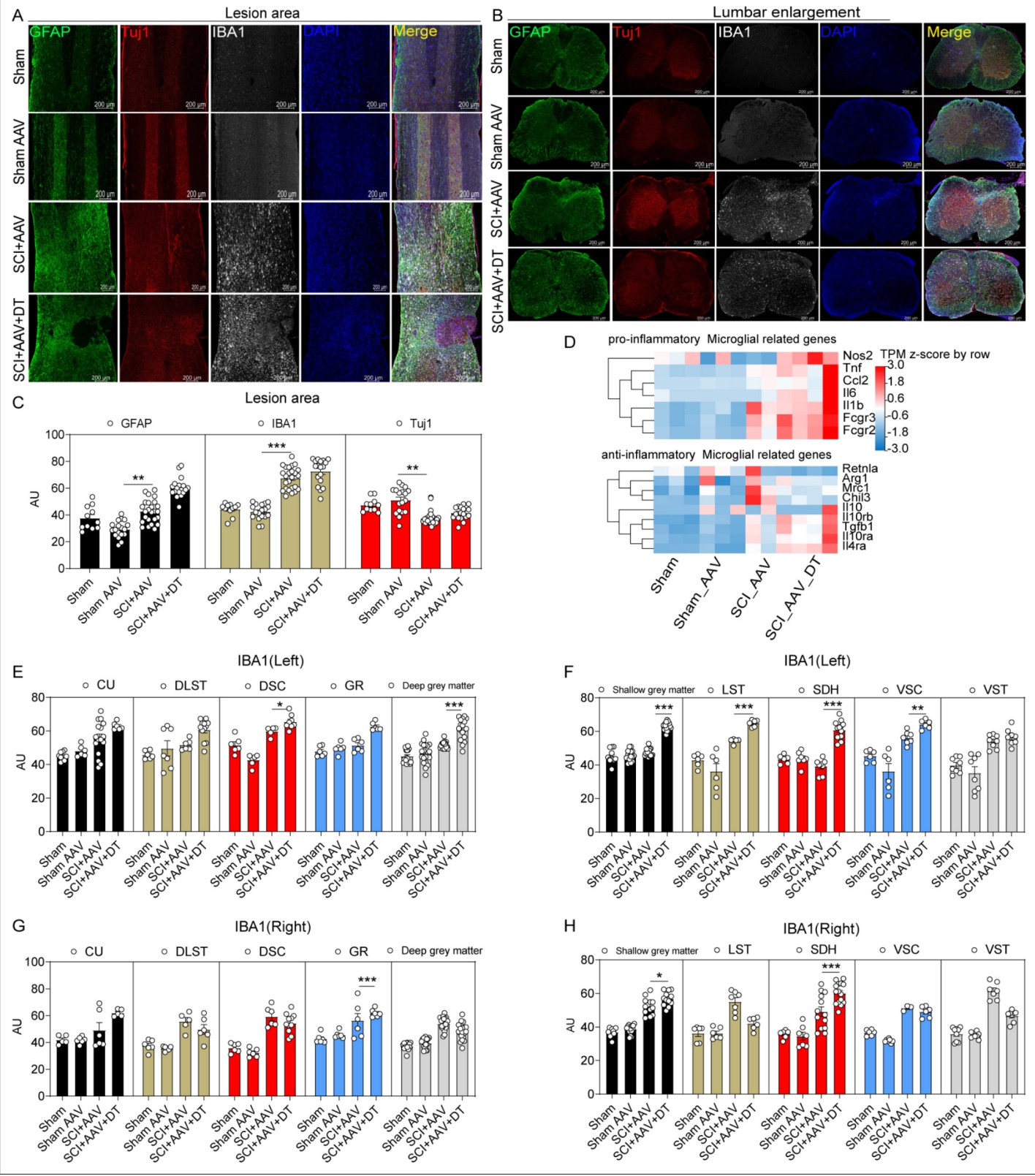

**Figure 4.** Microglia in lumbar enlargement were activated after selective astrocyte elimination. (**A–B**) Images of immunofluorescent staining using glial fibrillary acidic protein (GFAP) (green), Tuj1 (red), and IBA1 (white) as characteristic markers of astrocyte, neuron, and microglia, respectively. n=3 biological repeats. Scale bar, 200 μm. (**C**) The histogram and statistical results of GFAP, IBA1, and Tuj1 fluorescence intensity in the lesion area. (**D**) The heatmap showed that the transcripts of pro-inflammatory and anti-inflammatory microglial marker genes. n=3 biological repeats. (**E–H**) The histogram

*Figure 4 continued on next page*

*Figure 4 continued*

and statistical results of IBA1 fluorescence intensity in lumbar enlargement. CU = cuneate fasciculus, GR = gracile fasciculus, LST = lateral spinothalamic tract, DLST = dorsolateral spinothalamic tract, VST = ventral spinothalamic tract, DSC = dorsal spinocerebellar tract, VSC = ventral spinocerebellar tract, SDH = superficial dorsal horn. Values are the mean ± SEM. Statistical significance was determined by two-way ANOVA followed by the Student Newman–Keuls post hoc test. *$p<0.05$, **$p<0.01$, ***$p<0.001$, SCI + AAV + DT group vs. SCI + AAV group.

The online version of this article includes the following figure supplement(s) for figure 4:

**Figure supplement 1.** Full fluorescence images of the lesion area.

with the microglial marker IBA1 (*Figure 6E*). Thus, ISG15 and IRF7 were mainly expressed by microglia. To further confirm the expression of type I IFNs signaling factors were derived from microglia, the cultured and lipopolysaccharides (LPS)-activated microglia were used. Compared with the results for the control microglia (without LPS stimulation), the RT-PCR results revealed that 0.2 µg/mL LPS mainly activated pro-inflammatory microglia, including the pro-inflammatory microglial markers TNF-α, IL-1β, IL-6, and iNOS (*Figure 6G*). Anti-inflammatory microglial markers, including IL-4, TGF-β, IL-10, and Arg-1, were not significantly increased (*Figure 6G*). Additionally, RT-PCR results showed the upregulated mRNA expression of type I IFNs signaling critical genes in LPS-activated microglia, including *Sting1, Stat2, Stat1, Isg15, Irf9,* and *Irf3* (*Figure 6H*). These results indicated that type I IFNs signal was activated in microglia.

In order to further verify the activation of type I IFNs signal in microglia, different concentrations of interferon agonist (Ploy:IC) (*Khorooshi et al., 2015*) and STING agonist (ADU-S100 and DMXAA) (*Li et al., 2023*) were used to stimulate microglia at 12 and 24 hr. The mRNA expression of *Ifnb, Isg15, Stat2,* and *Stat1* increased significantly after Ploy:IC stimulation (*Figure 7A–B*). The mRNA expression levels of *Irf7, Ifnb, Stat2, Isg15, Stat1,* and *Irf9* significantly increased after stimulation with ADU-S100 and DMXAA (*Figure 7C–D*). A previous study identified that type I IFNs signals can significantly reduce sensory neuron activation and pain levels (*Donnelly et al., 2021*). Thus, the pain relief after selective resident astrocytes elimination may be associated with the activation of type I IFNs signal in microglia after SCI.

## Discussion

Astrogliosis is induced early in neuropathic pain models and sustained for exceptionally long durations, which covers both the acute-to-chronic pain transition and the maintenance phase of pain (*Ji et al., 2019*). Cytokines and chemokines produced by astrocytes can powerfully modulate excitatory and inhibitory synaptic transmission in pain circuits, leading to central sensitization, and a transition from acute to chronic neuropathic pain (*Ji et al., 2013*; *Ji et al., 2019*; *Kawasaki et al., 2008*). Recently, *Kohro et al., 2020* identified an SDH-localized Hes5 + astrocytes population that gated descending noradrenergic control of mechanosensory behaviour via α1A adrenergic receptors (α1A-Ars). Inhibition of locus coeruleus noradrenergic signal transduction to Hes5 + astrocytes could effectively prevent mechanical hyperalgesia induced by capsaicin (*Kohro et al., 2020*). Astrocytes activation increased in the lumbar enlargement of mice with SCI-induced pain, including the three largest ascending tracts and the SDH. These findings further highlight the importance of astrocytes in processing pain signals and targeting astrocytes may improve neuropathic pain after SCI.

Resident astrocytes maintain stable under a normal physiological state and rarely proliferates. SCI leads to a massive proliferation of resident astrocytes especially in the lesion area and re-express the neural progenitor cell markers, which are the most abundant and mainly located around the scar forming a region complementary to cells derived from ependymal cells (*Barnabé-Heider et al., 2010*; *Sabelström et al., 2014*). Ependyma is regarded as the main source of endogenous neural stem cells, which widely migrate to injury areas and generate the majority of neuroprotective astrocytes after SCI (*Barnabé-Heider et al., 2010*; *Meletis et al., 2008*). However, only a small number of ependymal cells could proliferate and migrate to the lesion score after SCI (*Ren et al., 2017*). Transgenic mice and lineage tracing found that ependymal cells did not produce a large number of astrocytes. Instead, ependymal cells differentiated to Tuj1 positive neurons in the epicenter. NG2 cells were previously considered to be the component of glial scar after SCI (*Church et al., 2016*; *Hackett and Lee, 2016*; *Hesp et al., 2015*). However, reports about their functions are still controversial, partly because there are many cell types that up regulate proteoglycan NG2 in the scar and lesion core after SCI (*Nishiyama*

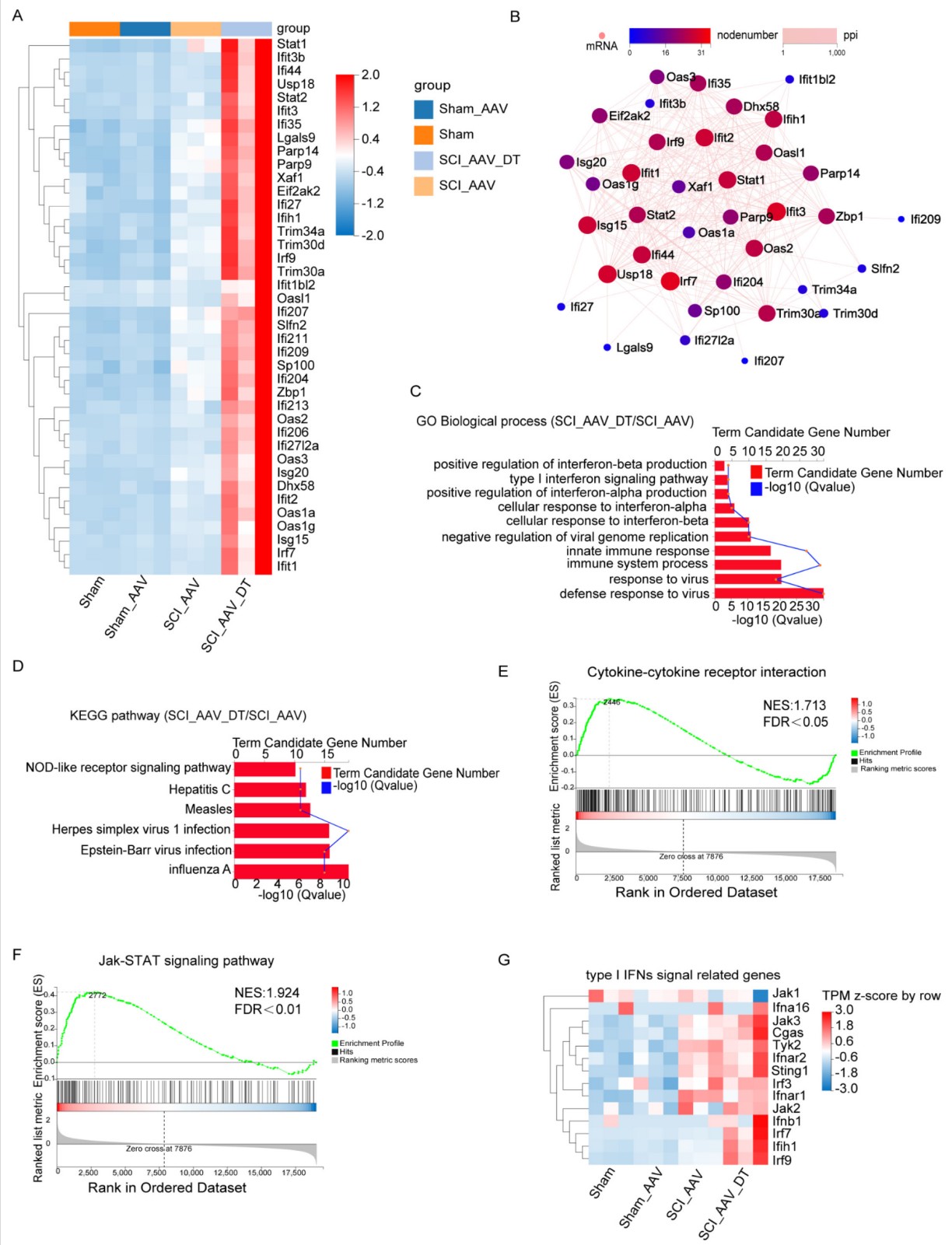

**Figure 5.** Selective astrocyte elimination activated Type I interferons (IFNs) signal. (**A**) The heatmap showed the differentially expressed genes (DEGs) between the SCI + AAV + DT group and the SCI + AAV group. (**B**) Protein-protein interaction (PPI) network analysis showed the interaction of the DEGs between the SCI + AAV + DT group and the SCI + AAV group. (**C**) Gene ontology (GO) enrichment analysis revealed the dominant biological process of DEGs between the SCI + AAV + DT group and the SCI + AAV group. (**D**), Kyoto Encyclopedia of Genes and Genomes (KEGG) pathway showed

*Figure 5 continued on next page*

*Figure 5 continued*

dysregulated signaling pathways between the SCI + AAV + DT group and the SCI + AAV group. (**E–F**) Gene set enrichment analysis (GSEA) analysis showed the dysregulated signaling pathways between the SCI + AAV + DT group and the SCI + AAV group. (**G**) The heatmap showed that the critical gene expression involved in type I IFNs signal. n=3 biological repeats.

The online version of this article includes the following figure supplement(s) for figure 5:

**Figure supplement 1.** RNA-seq analysis results between the SCI + AAV group and sham adeno-associated virus (AAV) group.

*et al., 2005*). Totally, through lineage tracing, these findings determined that the resident astrocytes but not ependymal cells were the main origins of astrocytes-induced neuropathic pain after SCI.

One of the novel findings was that selective resident astrocytes elimination in the lumbar enlargement could attenuate neuropathic pain after SCI. Previously, there was limited data on the association between local astrocytes depletion and neuropathic pain attenuation. Astrocytes were targeted astrocytes in the spinal cord by using GFAP-Cre; *Gt(ROSA)26Sor^em1(CAG-LSL -tdTomato-2A-DTR)Smoc* mice, but the mice were not suitable for labelling astrocytes (*Figure 2—figure supplement 1*). The most reliable mouse strain for targeting astrocytes in vivo is Aldh1l1-Cre/ERT2 mice (*Srinivasan et al., 2016*). To selectively and locally target astrocytes in the spinal cord, instead, selective astrocytes elimination in lumbar enlargement was performed by AAV2/5-GfaABC1D-Cre mediated local astrocytes transduction and DT injection. Several previous studies have reported astrocytes elimination in animal experiments. Through the transgenical method, a previous study found that domain-specific depletion of astrocytes in the dorsal spinal cord showed atrophy, reduction in the total number of astrocytes, loss of neuropil, and congested neurons (*Tsai et al., 2012*). However, they did not observe increased inflammation, gliosis, or blood-brain barrier (BBB) permeability in these mice (*Tsai et al., 2012*). They speculated that the remaining astrocytes were sufficient for structural maintenance (*Tsai et al., 2012*). In contrast, domain-specific depletion of astrocytes in the ventral spinal cord resulted in abnormal motor neuron synaptogenesis, which was not rescued by the immigration of astrocytes from adjoining regions (*Tsai et al., 2012*). With transgenically targeted DTR and AAV2/5-GfaABC1D-Cre injection, one another study found that DT-mediated ablation of chronic astrocytic scar could significantly prevent axon regrowth after severe SCI in mice (*Anderson et al., 2016*). Two other studies also found that astrocytes elimination of the whole body impaired redox homeostasis and triggered neuronal loss in the central nervous system, accompanied by myelin sheath loss and motor function impairment (*Allnoch et al., 2019*; *Schreiner et al., 2015*). Meanwhile, local astrocytes depletion through local DT injection did not affect astrocytes in other sites as shown in their supplemental information (*Schreiner et al., 2015*). These results also showed that selective resident astrocytes elimination in the lumbar enlargement does not have a significant impact on weight and motor function.

The transcripts of pro-inflammatory microglial marker genes elevated and microglia could activate after selective resident astrocytes elimination. As the resident macrophages in the spinal cord, microglia play critical roles in the progress of neuropathic pain (*Chen et al., 2018*; *Salter and Stevens, 2017*). However, the role of microglia in chronic inflammatory pain is still controversial (*Chen et al., 2018*). A previous study reported that P2x4 knockout mice exhibit unaltered inflammatory pain (*Tsuda et al., 2009*). In contrast, Cx3cr1 knockout mice exhibit reduced inflammatory pain (*Staniland et al., 2010*). Recently, *Tansley et al., 2022* found that after peripheral nerve injury, microglia will degrade the extracellular matrix structure (neuronal peripheral network, PNN) in lamina I of the spinal dorsal horn, enhance the activity of projection neurons, and induce pain-related behaviors. By comparison, *Kohno et al., 2022* described a spinal cord–resident pool of CD11c[+] microglia that emerges during pain maintenance and contributes to the resolution of neuropathic pain in mice. Interestingly, previous evidence indicated that direct activation of astrocytes using an optogenetic approach induces microglial reactivity and pain hypersensitivity (*Nam et al., 2016*). Thus, microglia act as diverse roles in a pain development and resolution (*Sideris-Lampretsas and Malcangio, 2022*). Sex differences in the involvement of microglia in mediating chronic pain have been demonstrated (*Sorge et al., 2015*; *Taves et al., 2016*). The effect of sex on neuropathic pain was also explored, and observed no significant differences between male and female mice. Thus, there were inconsistent associations between sexes and neuropathic pain after SCI (*Dominguez et al., 2012*; *Mapplebeck et al., 2016*; *McFarlane et al., 2020*). Furthermore, the onset of cold/thermal hypersensitivity were delayed than mechanical allodynia, which may be related to the different processing of different types of pain signals.

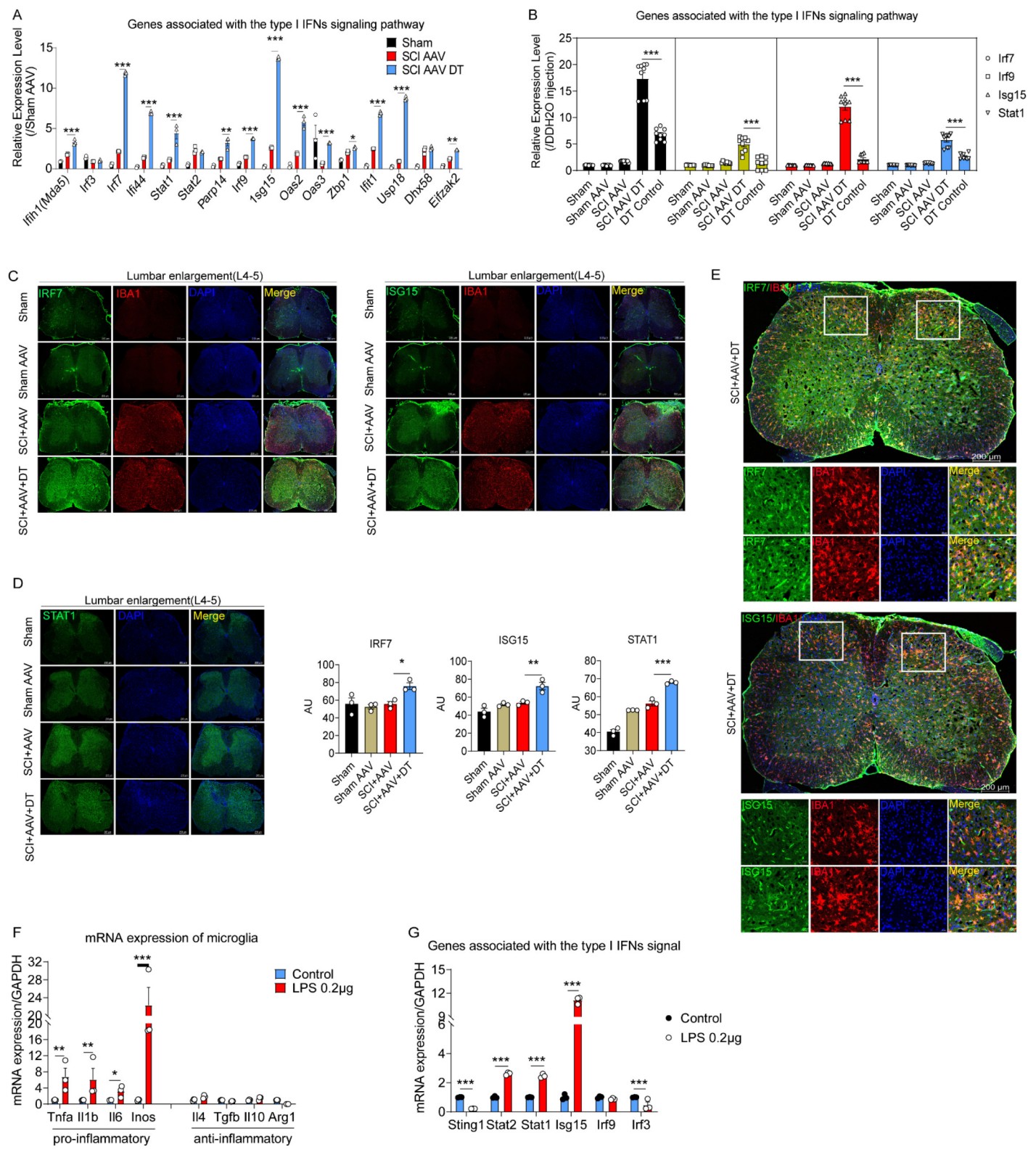

**Figure 6.** The type I interferons (IFNs) signal was activated in microglia after selective resident astrocyte elimination. (**A–B**) RT-PCR identified the effects of selective astrocyte elimination and Diphtheria toxin (DT) injection alone on mRNA expression of type I IFNs signal genes. n=3 biological repeats. *p<0.05, **p<0.01, ***p<0.001. (**C–D**) Images of immunofluorescent staining and statistical results using IRF7, ISG15, and STAT1 as critical proteins of type I IFNs signal and microglia marker IBA1. Scale bar, 200 μm. n=3 biological repeats. *p<0.05, **p<0.01, ***p<0.001, SCI + AAV + DT group vs.

*Figure 6 continued on next page*

*Figure 6 continued*

SCI + AAV group. (**E**) Images of immunofluorescent staining using IRF7, ISG15, and IBA1. Scale bar, 200 μm. n=3 biological repeats. (**F–G**) The mRNA expression of pro- and anti-inflammatory microglial markers and type I IFNs signal critical genes after 0.2 μg/mL lipopolysaccharides (LPS) stimulation. n=3 biological repeats. *p<0.05, **p<0.01, ***p<0.001, Control group vs. LPS 0.2 μg group. All values are the mean ± SEM. Statistical significance was determined by two-way ANOVA followed by the Student Newman–Keuls post hoc test.

Among the most significant DEGs after selective resident astrocytes elimination were the genes involved in regulating type I IFNs production, and the signal transduction of type I IFNs was significantly upregulated. Molecular experiment results also found the mRNA expression of genes associated with the type I IFNs signal increased after selective astrocytes elimination. It has been verified that type I IFNs signal can significantly reduce sensory neuron activation and pain levels (***Donnelly et al., 2021***). Mice with type I IFNs signal deficiency were more sensitive to pain stimuli and had more obvious activation of intense sensory neurons (***Donnelly et al., 2021***). Thus, type I IFNs signal might be involved in the association between selective resident astrocytes elimination and neuropathic pain attenuation. Although interferons could derive from various cells, myeloid-lineage cells, and microglia are resident immune cells in the central nervous system and are believed to be the main source of IFN-β within the CNS during inflammatory conditions (***Owens et al., 2014***). Reinert and colleagues

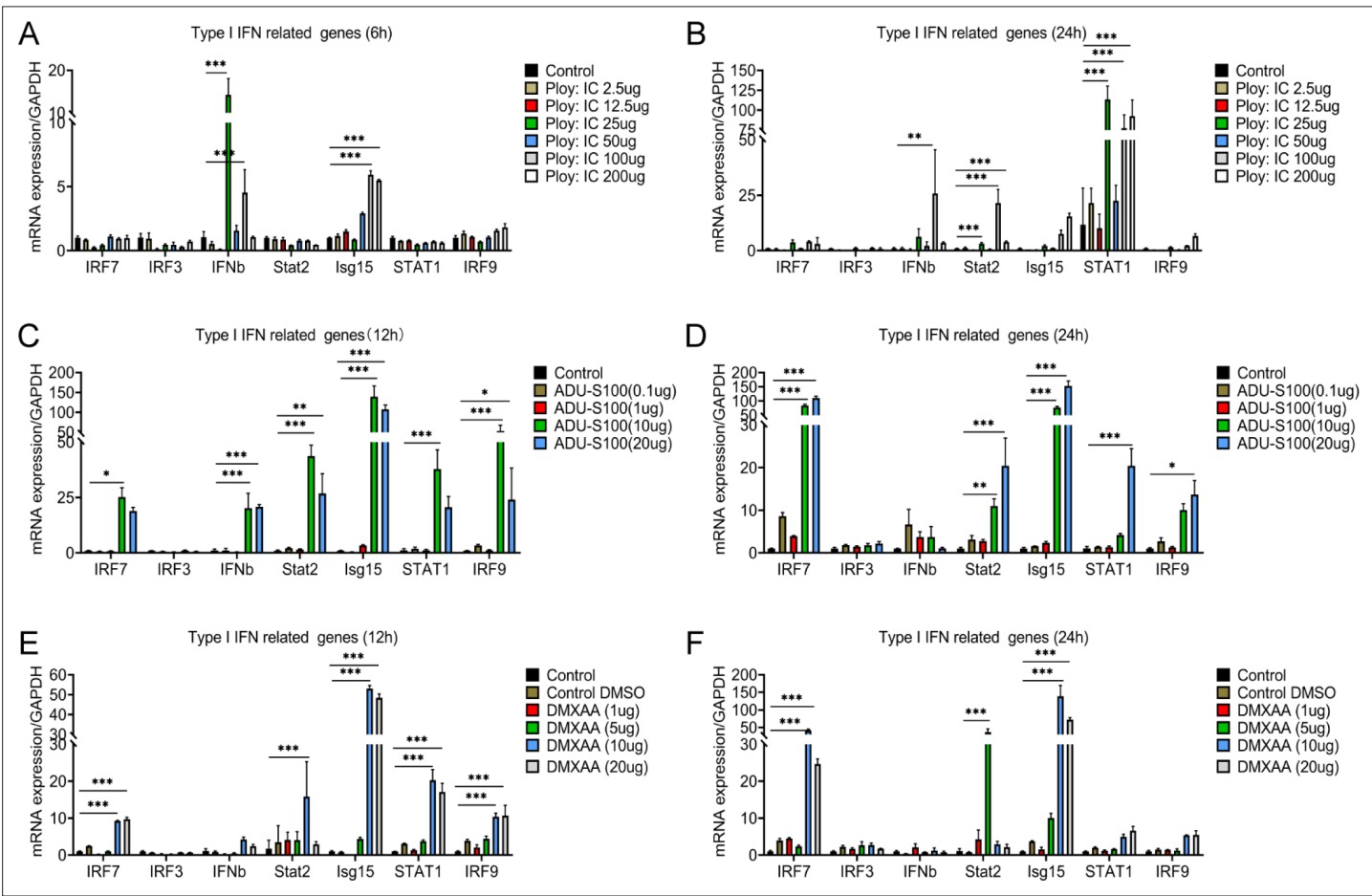

**Figure 7.** Interferon agonist (Ploy:IC) and an STING agonist (ADU-S100 and DMXAA) significantly activated the type I interferons (IFNs) signal in microglia. (**A-F**) The mRNA expression levels of *Irf7, Irf7, Ifnb, Stat2, Isg15, Stat1,* and *Irf9* after stimulating with different concentrations of Ploy:IC, ADU-S100, and DMXAA. n=3 biological repeats. *p<0.05, **p<0.01, ***p<0.001, treatment group vs. control group. All values are the mean ± SEM. Statistical significance was determined by two-way ANOVA followed by the Student Newman–Keuls post hoc test.

The online version of this article includes the following figure supplement(s) for figure 7:

**Figure supplement 1.** The potential schematic diagram of selective resident astrocytes elimination attenuated neuropathic pain after spinal cord injury (SCI).

(*Reinert et al., 2016*) identified microglia as the primary producers of type I IFNs, and the expression of cGAS and Sting1 were enriched in microglia (*Chin, 2019*). The association between selective astrocytes elimination, type I IFNs signal, and microglia activation remains unclear. Considering that the death or ablation of numerous cell types in CNS including oligodendrocytes could spark the activation of microglia (*Gritsch et al., 2014*), more studies were needed to explore the effect of microglia activation on the association between astrocytes elimination and neuropathic pain attenuation.

There are also some limitations. Although these results determined that the selective resident elimination could significantly attenuate neuropathic pain, astrocyte elimination is difficult to achieve in clinical practice. Instead, these findings provided a strategy to potentially target and activate type I IFNs signals to improve pain after SCI. Additionally, astrocytes elimination may cause many pathophysiological changes in addition to activating type I IFNs signal, the observed effect may be a side effect of the elimination of astrocytes. For these experiments did not identify whether diphtheria toxin-induced cell apoptosis leads to the activation of the type I IFNs signaling pathway in microglia and is not induced only by astrocyte-specific clearance. These experiments in cell culture induced by LPS were partly to give a possible mechanist explanation, the optimal methods should be performed in primary microglia obtained from the SCI + AAV group or SCI + AAV + DT group and compared to shame operated one. Finally, these results also did not provide the rescue experiment, to explore the homeostatic function of astrocytes and the roles of interferon agonists (Ploy:IC) and STING agonists (ADU-S100 and DMXAA).

## Conclusion

Above all, these findings determined that resident astrocytes but not ependymal cells are the main origins of astrocytes participate in neuropathic pain after SCI. The selective resident elimination in lumbar enlargement could significantly attenuate neuropathic pain, and the mechanisms were partly associated with type I IFNs signal activation in microglia (as the schematic diagram shown in *Figure 7—figure supplement 1*). Therefore, targeted intervention on type I IFNs signal may be an effective strategy for neuropathic pain treatment after SCI.

# Materials and methods
## Experimental models (organisms/strains)

The experimental procedures were designed to minimize the number of animals used as well as animal suffering. All animal experiments were carried out in accordance with the U.K. Animals (Scientific Procedures) Act, 1986 and associated guidelines, EU Directive 2010/63/EU for animal experiments, and approved by the Institutional Animal Care and Use Committee (IACUC) of the Tongji University School of Medicine (SYXK2019-0005). All transgenic and non-transgenic mice used were derived from in-house breeding colonies maintained on C57/BL6 backgrounds. Mice were housed on a 12/12 hr light/dark cycle and had food and water available ad libitum. The room temperature and humidity were appropriate. All behavioral testing was performed between 9:00 am and 5:00 pm. Transgenic mice were purchased from the Model Organisms Centre, Shanghai, and the Jackson Laboratory. To target spinal cord ependymal cells, C57BL/6-*Foxj1*$^{em1(GFP-CreERT2-polyA)Smoc}$ mice (Cat# NM-KI-200133, the Model Organisms Centre) were used in this experiment. C57BL/6-*Foxj1*$^{em1(GFP-CreERT2-polyA)Smoc}$ mice were under the control of promoters with specific expression in ependymal cells (FoxJ1), which have been widely used for identification of cells that express Foxj1 in the central nervous system and peripheral organs (*Barnabé-Heider et al., 2010*; *Ren et al., 2017*). Gt(ROSA)26Sor$^{em1(CAG-LSL-tdTomato-2A-DTR)Smoc}$ mice were obtained from *Gt(ROSA)26Sor*$^{em1(CAG-LSL-RSR-tdTomato-2A-DTR)Smoc}$ mice (Cat# NM-KI-190086, the Model Organisms Centre) crossed with C57BL/6J$^{Smoc-Tg(CAG-Dre)Smoc}$ (Cat# NM-TG-00026, the Model Organisms Centre). *Gt(ROSA)26Sor*$^{em1(CAG-LSL -tdTomato-2A-DTR)Smoc}$ mice carried the reporter gene tdTomato and diphtheria toxin receptor (DTR), which can be used for lineage tracing and selective elimination for specific cell population (*Buch et al., 2005*; *Madisen et al., 2010*), including astrocytes (*Anderson et al., 2016*). GFAP-Cre; *Gt(ROSA)26Sor*$^{em1(CAG-LSL -tdTomato-2A-DTR)Smoc}$ mice were obtained from *Gt(ROSA)26Sor*$^{em1(CAG-LSL- tdTomato-2A-DTR)Smoc}$ mice crossed with FVB-Tg$^{(GFAP-cre)25Mes/J}$ mice (Cat# JAX004600, the Jackson Laboratory). Wild-type C57BL/6 J mice were obtained through reproduction. For genotyping, genomic DNA was isolated from ear biopsy for PCR. Transgenic mice and genotyping primers were listed in *Supplementary file 1a*. B6.Cg-*Gt(ROSA) 26Sor*$^{tm9(CAG-tdTomato) Hze}$/J mice (Cat# JAX007909,

the Jackson Laboratory) were used for the AAV2/5-GfaABC1D-Cre test experiment. A total of 132 *Gt(ROSA)26Sor*$^{em1(CAG-LSL-tdTomato-2A-DTR)Smoc}$ mice aged 2 mo and older were randomly and equally divided into the following groups: the sham group (male = 15, female = 15), sham AAV group (male = 15, female = 15), SCI +AAV group (male = 18, female = 18), and SCI + AAV + DT group (male = 18, female = 18). Three or more independent experiments were performed.

## Spinal cord contusion model and motor function test

Thoracic spinal cord contusion, a widely used SCI model for neuropathic pain (*Kramer et al., 2017*), and the MASCIS Impactor Model III (W.M. Keck Center for Collaborative Neuroscience, Rutgers, the State University of New Jersey, USA) were applied (*Duan et al., 2021*). Briefly, mice were weighed and deeply anaesthetized with isoflurane evaporated in a gas mixture containing 70% $N_2O$ and 30% $O_2$ through a nose mask. The back skin was shaved and cleaned with 75% alcohol. A laminectomy at T10 was performed to remove the part of the vertebra overlying the spinal cord, exposing a circle of dura through an operating microscope (Zeiss, Germany) and rodent stereotaxic apparatus (RWD Life Science Co., Ltd, Shenzhen, China). The spinal column was stabilized using lateral clamps over the lateral processes at T9 and T11. Contusion was performed at T10 with a 5 g impactor and 6.25 mm height with a force of about 60kdyn, which could cause a moderate injury as previously reported (*Wu et al., 2013*; *Wu et al., 2014*), and then the wound was sutured. Mice had received natural illumination to keep warm before and after the surgery. The following symbols were indicators of a successful contusion model: (1) the impact point was located in the middle of T10, (2) paralysis of both hindlimbs occurred after awakening, (3) the cut-offs of mechanical pain were lower than their baselines. Unsuccessful models were excluded in the following experiment and analysis. Sham mice underwent laminectomy but not contusion. Urine was manually expressed from the bladders of the injured mice twice per day until autonomous urination recovered. Hindlimb motor function was tested in an open field chamber on days 0, 1, and 3 after SCI and weekly for up to 6 wk. The Basso Mouse Scale (BMS) (*Basso et al., 2006*) was applied to quantify motor function by two investigators blinded to the genotypes and group assignments.

## Measurement of mechanical sensitivity with an electronic von Frey apparatus

The mechanical allodynia of the hindlimbs was measured by the electronic von Frey apparatus (IITC Life Sciences, Woodland Hills, CA) as previously reported (*Cunha et al., 2004*; *Martinov et al., 2013*), which is a sensitive, objective, and quantitative measurement tool for mechanical sensitivity in rodent models with pain (*Cunha et al., 2004*; *Martinov et al., 2013*). The electronic von Frey apparatus has a pressure-meter which consist of a hand-held force transducer fitted with a 0.5 mm$^2$ polypropylene tip (*Cunha et al., 2004*). When a positive response appeared, the intensity of the stimulus was automatically recorded. The investigators were proficiency trained in advance to use the polypropylene tip vertically to the central of the plantar. Briefly, mice were placed on a wire mesh platform in individual Plexiglas cubicles and allowed to acclimate to the Plexiglas restraint and testing room for 30 min. Before testing, the mice were quiet without exploratory behaviors or defecation, and not resting on their paws (*Cunha et al., 2004*). The average interstimulus interval was 2 min. Polypropylene tip was applied a vertical and gradual increase force to the plantar surface of the left and right hindlimbs for 3–4 s, or until a positive response appeared. Behaviours that were considered positive responses to the filament included brisk paw withdrawal, flinching, hunching of the back, licking of the stimulated area, and escape responses (*Wu et al., 2016*). All animals underwent two trials (five measurements/trial) at 24 and 48 hr before contusion to calculate the average plantar paw withdrawal threshold of both hindlimbs at baseline. The measurement of post-contusion withdrawal thresholds was performed in a blinded fashion by two independent investigators at 2, 3, 4, 5, and 6 wk after contusion. These timepoints were chosen for the nociceptive assays to calculate BMS scores was to avoid the issue of motor impairment for reflex-based assays, and these mice could maintain a standing posture from 2 wk after contusion on.

## Analgesia test for thresholds of cold and thermal pain

Cold and hot plate (Cat# YSL-21; Shanghai Yuyan Instruments Co., Ltd. China) was used to assess cold and thermal pain as previously reported (*Carozzi et al., 2013*; *Wu et al., 2016*). Cold and thermal pain

tests of the hindlimbs were conducted 1 d after mechanical allodynia measurement. Before testing, mice were also transferred into the testing room to acclimate to the environment for 30 min. The temperatures of 0 °C (cold stimulus) and 50 °C (hot stimulus) were used to measure the thresholds for noxious heat and cold, respectively (*Wu et al., 2016*). To avoid tissue injury, temperatures of 0 °C (cold stimulus) for 5 min and 50 °C (hot stimulus) for 30 s were set as the cut-off points. The positive response included licking, jumping, or hind paw withdrawal. The paw withdrawal latency (for hot stimuli) and paw withdrawal frequency (for cold stimuli) of positive responses were recorded. For hot stimuli, the paw withdrawal latency means the period from touching the hot plate to the emergence of a positive response. For cold stimuli, the paw withdrawal frequency means the number of nociceptive responses observed in 5 min on a cold plate. The tests were carried out 24 hr before contusion and 2, 3, 4, 5, and 6 wk after contusion. The analgesia test for thresholds of cold and thermal pain were conducted by two investigators blinded to the groups and experimental conditions.

## AAV2/5-GfaABC1D-Cre and DT injection

AAV2/5 with the GfaABC1D promoter (AAV2/5-GfaABC1D-Cre) was used to target Cre recombinase expression exclusively in astrocytes (*Anderson et al., 2016*). The 681 bp GFAP promoter, GfaABC1D, was derived from the 2.2 kb human GFAP promoter gfa2. Importantly, the GfaABC1D promoter is about twice as active as gfa2 with a smaller size, which is better suited to viral vectors (*Lee et al., 2008*; *Yu et al., 2020*). The expression of cargo genes under the GfaABC1D promoter in astrocytes is highly efficient and specific (*Yu et al., 2020*). Previous studies reported that AAV2/5 predominantly labeled astrocytes, displaying robust co-localization of reporter gene (eGFP) with the astrocytic marker, GFAP (*Ortinski et al., 2010*). The AAV2/5 vector with a minimal GFAP promoter (AAV2/5-GfaABC1D-Cre) has been regarded as the most reliable viral approach and widely used to target astrocytes (*Yu et al., 2020*). AAV2/5-GfaABC1D-Cre was purchased from Shanghai Taitool Bioscience Co., Ltd. (Cat# S0611−5, ≥1E+13 V.G/ml). AAV2/5-GfaABC1D-Cre was injected into $Gt(ROSA)26Sor^{em1(CAG-LSL--tdTomato-2A-DTR)Smoc}$ mice of SCI + AAV group and SCI + AAV + DT group 2 wk before T10 contusion, and into mice of sham AAV group 2 wk before T10 laminectomy. Sham group mice underwent L4-5 laminectomy but not AAV. Briefly, after precise location of L4-5 spinal cord as previously reported (*Rigaud et al., 2008*), the back skin was shaved and cleaned with 75% alcohol. A laminectomy was performed at L4-5 as described above. AAV2/5-GfaABC1D-Cre was injected into the both sides of a dorsal median of the lumbar enlargement (0.5 mm lateral to midline with a depth of 0.6 mm, 0.4 μL per side, ≥1E+11 V.G/ml in sterile saline), with a Hamilton 33-gauge (33 G) microinjection needle at a speed of 0.2 μL/min. After injection, the needle was held for 5 min to avoid liquid reflux. Overall, no obvious tissue injury of a spinal cord, and no motor and sensory function impairment of hindlimbs were observed during the following 2 wk. Spinal cord tissue fluorescence images were taken and handled with a Fluorescence-labeled Organism Bioimaging Instrument (FOBI, Neoscience Ltd., Suwon City, Rep. of Korea) after AAV2/5-GfaABC1D-Cre injection.

## DT injection

Mice received 500 ng diphtheria toxin (DT, Cat# 150, List Biological Laboratory) in 100 μL distilled, deionized water (ddH$_2$O) through intraperitoneal injection (i.p.) (*Binnewies et al., 2019*) on 31, 32, and 34 dpi (days post-injury), when the pain level peaked (the lowest paw withdrawal threshold of mechanical sensitivity). Then, the mice were sacrificed at 1 wk after DT injection for RNA sequencing, bioinformatics analysis, immunohistochemistry, and molecular experiment.

## RNA sequencing (RNA-seq) and bioinformatics analysis

To further evaluate the role of selective astrocytes elimination in gene expression regulation and pain attenuation mechanisms, RNA-seq was used to examine the transcriptomes of lumbar enlargement tissues in the sham group (n=3), sham AAV group (n=3), SCI + AAV group (n=3), and SCI + AAV + DT group (n=3). Previous findings revealed the sex-dependent difference in neuropathic pain after SCI (*Fiore et al., 2022*; *Stewart et al., 2021*). Thus, the lumbar enlargement tissues of female mice were selected for the RNA-seq experiment. Briefly, mice were euthanized after terminal anaesthesia by pentobarbital overdose. Spinal cord tissue was dissected from lumbar enlargements (L4-5) at 1 wk after selective astrocytes elimination, with a dissecting microscope, and 2 mm segments were harvested and marked. A total of 12 samples were sent to the Beijing Genomics Institute (BGI)

Company (Shenzhen, China) in solid carbon dioxide for further RNA-seq analysis, and sequencing was performed on a DNBSEQ platform at BGI Company. Bioinformatics analysis, including GO enrichment, Kyoto Encyclopedia of Genes and Genomes (KEGG) pathway, GSEA, and PPI network analysis, was carried out with the online platform Dr. Tom (BGI Company). Only genes with transcripts per million (TPM) >1 were analyzed. Differentially expressed genes (DEGs) were identified using the DEGseq method and screened with the criteria of q value ≤0.05 and $\log_2 FC$ ≥0.6, where FC represents the fold change. False discovery rate (FDR) method was used to adjust for multiple comparisons to generate q-values in the RNA-seq study. The normalized enrichment score (NES) was utilized to select the most activated and inactivated GO terms and pathways. The sequence and sample data have been deposited in the NCBI database under Sequence Read Archive (SRA) with Bioproject identification number PRJNA847704 (Accession number: SRR19611667 - SRR19611678). The single-cell RNA sequencing (scRNA-seq) results were analyzed through online data from injured mouse spinal cords (https://jaeleelab.shinyapps.io/sci_singlecell/) (*Milich et al., 2021*).

## Immunofluorescence staining and analysis

Immunofluorescence staining procedures were conducted as described previously (*Ren et al., 2017*). After post-fixation and cryoprotection, a dissected 6 mm segment of spinal cord centred around the injury epicentre, or a 2 mm segment of the lumbar enlargement (L4-5), was coronally sectioned at 12 μm thickness and thaw-mounted onto Superfrost Plus slides (Citotest Labware Manufacturing Co., Ltd.). The primary antibodies used were as follows: glial fibrillary acidic protein (GFAP, Cat# ab4674, Abcam, 1:500), Neuronal nuclear antigen (NEUN, Cat# ab190565, Abcam, 1:500), Tubulin beta III (Tuj1, Cat# MAB1637, Millipore, 1:500), Ionized calcium-binding adaptor molecule 1 (IBA1, Cat# 016–20001, Wako, 1:500), Oligodendrocyte Transcription Factor 2 (Olig2, Cat# AF2418, R&D system, 1:500), S100 Calcium Binding Protein A10 (S100A10, Cat#11250–1-AP, Protein Tech, 1:250), Interferon-stimulated Gene 15 (ISG15, Cat# sc-166755, Santa Cruz, 1:50), Interferon regulatory factor 7 (IRF7, Cat# sc-74471, Santa Cruz, 1:50), Signal transducer and activator of transcription 1 (STAT1, Cat# ab109461, Abcam, 1:150). The secondary antibodies were Alexa Fluor 488 (Cat# abs20019A, Absin, 1:500), Alexa Fluor 488 (Cat# A-32814, Invitrogen, 1:500), Alexa Fluor 488 (Cat# A-32766, Invitrogen, 1:500), Alexa Fluor 555 (Cat# A-32773, Invitrogen, 1:500), Alexa Fluor 594 (Cat# ab150176, Abcam, 1:500), and Alexa Fluor 647 (Cat# A-32795, Invitrogen, 1:500). The nuclei were stained with 4',6-diamidino-2-phenylindole (DAPI, Cat# C1002, Beyotime Institute of Biotechnology), and fluorescence images were taken by confocal microscopy (LSM 700, Carl Zeiss, Jena, Germany). Image acquisition was performed with ZEN 2.3 (blue edition, Carl Zeiss), and micrographs were assembled using Adobe Illustrator CC 2018. As previously reported (*Watson, 2012*), we selected four regions in the lumbar enlargement to analyze in the immunohistochemistry images, including the three largest ascending tracts and SDH. The three largest ascending tracts were (1) the gracile and cuneate fasciculi (GR and CU, respectively); (2) the lateral, dorsolateral, and ventral spinothalamic tracts (LST, DLST, and VST, respectively); and (3) the dorsal and ventral spinocerebellar tracts (DSC and VSC, respectively). These tracts are closely associated with proprioception and sensation (*Watson, 2012*). For immunofluorescence analysis of lumbar enlargement, three sections were randomly selected to delineate the three tracts and SDH (three areas per tract/SDH), then the 160 μm × 160 μm images were exported as single channel images through ZEN 2.3 (blue edition). Lesion area covered the lesion core, rostral, and caudal injured spinal cords at a distance of 500 μm from the lesion core, respectively. A total of three sections (eight areas of rostral and eight of caudal injured spinal cords per section) were randomly selected per mice. Then the 160 μm × 160 μm images were exported by ZEN 2.3 (blue edition). Image-J software with customized macros was used to quantify the fluorescence intensity of proteins including the GFAP, Tuj1, IBA1, ISG15, IRF7, and STAT1 in the images mentioned above. For the sham group and treatment groups, three mice per group were tested.

## Microglial culture

To observe the production of type I interferon-related factors by microglia after activation, lipopolysaccharides (LPS, Cat# *Escherichia coli* O111:B4, Sigma-Aldrich) was used activated microglia induced Furthermore, different concentrations of interferon agonist (Ploy:IC (Cat# B5551, APExBIO)) (*Khorooshi et al., 2015*) and Stimulator of interferon genes (STING) agonist (ADU-S100 (Cat# HY-12885B, MCE) and DMXAA (Cat# HY-10964, MCE)) (*Li et al., 2023*) were used to stimulate microglia

for 12 and 24 hr. Mouse microglial SIM-A9 cells were purchased from ATCC (Cat# CRL-3265, ATCC) and cultured according to ATCC recommended growth conditions in DMEM-F12 (Cat# 11330057, Gibco) containing 10% heat-inactivated FBS (Cat# 16140–071, Gibco) and 5% heat-inactivated horse serum (Cat# 26050–088, Gibco) (*Correia et al., 2021*). LPS at concentrations of 0.2 µg/mL LPS (*Jayakumar et al., 2022*) was used to stimulate SIM-A9 cells after they reached 60–70% confluency for 24 hr. Microglia and culture medium were collected and centrifuged at 12,000 × g for 5 min to remove debris. The expression of pro-inflammatory and anti-inflammatory microglial marker genes were measured by RT-PCR. After the stimulation by Ploy:IC (2.5 µg, 12.5 µg, 25 µg, 50 µg, 100 µg, 200 µg,), ADU-S100 (0.1 µg, 1 µg, 10 µg, 20 µg) and DMXAA (1 µg, 5 µg, 10 µg, 20 µg) for 12 and 24 hr, the microglia were also obtained for RT-PCR experiment. Control wells were also established, and the data were analyzed and compared with those of the control wells.

## Quantitative real-time PCR (RT–PCR)

The total RNA of cells was isolated with RNAiso plus (Cat# 9108, Takara). The concentration and purity of RNA samples were measured using a Nanodrop ND-2000 (Thermo Science, MA, USA) for further experiments. Five hundred nanograms of RNA was converted to complementary DNA (cDNA), which was synthesized with a PrimeScript reverse transcriptase kit (Cat# RR037A, Takara). RT–PCR was performed using a TB Green TM Premix Ex Taq Kit (Cat# RR820A, Takara) on a Light Cycler Real-Time PCR System (480II, Roche). The primer sequences (Shanghai Generay Biotech Co., Ltd.) were designed through PrimerBank (https://pga.mgh.harvard.edu/primerbank/) and are listed in *Supplementary file 1b-d*. The relative amounts of mRNA were calculated using the ΔΔCt relative quantification method. *Gapdh* served as the control gene, and the mRNA levels of specific genes were normalized to *Gapdh*. Calculations and statistics were performed in Microsoft Excel version 16.36. Graphs were plotted in GraphPad Prism 8 version 8.4.3.

## Statistical analysis

All experiments were conducted with three or more duplicates. All continuous data were shown as mean ± SEM. Two-way ANOVA was performed followed by Student Newman–Keuls post hoc test for continuous data. p-values <0.05 were considered statistically significant. Data analyses were conducted using the Statistical Analysis System (SAS), version 9.4 (SAS Institute, Inc, Cary, NC, USA). Plots were generated using GraphPad Prism 8 software (GraphPad Software, San Diego, CA).

## Acknowledgements

These experiments were supported by the International Cooperation Project of the National Natural Science Foundation of China (Grant No. 81810001048), the National Natural Science Foundation of China (Grant Nos. 81974190, 82271419, 82225027, 81901902, and 81701217), the National Key R&D Program of China (Grant No. 2020YFC2008703), Shanghai Rising-Star Program (Grant No. 22QA1408200), the Fundamental Research Funds for the Central Universities (22120220555 and 22120230138), Programs in Emergency and Critical Care Medicine, Shanghai Municipal Health Commission (KPB1702), National Key Specialty Program, National Health Commission of China (GJ2301). We thank Gufa Lin, Bei Ma, Zhourui Wu and Xu Xu of our laboratory for help and advice. We also thank Pianpian Fan (Department of Pediatrics, West China second hospital, Sichuan University) for article grammar check and suggestions.

## Additional information

### Funding

| Funder | Grant reference number | Author |
| --- | --- | --- |
| International cooperation project of National Natural Science Foundation of China | 81810001048 | Liming Cheng |

| Funder | Grant reference number | Author |
|---|---|---|
| Natural Science Foundation of China | 81974190 | Ning Xie |
| Natural Science Foundation of China | 82271419 | Yanjing Zhu |
| Natural Science Foundation of China | 82225027 | Rongrong Zhu |
| Natural Science Foundation of China | 81901902 | Yanjing Zhu |
| Natural Science Foundation of China | 81701217 | Yilong Ren |
| National Key R&D Program of China | 2020YFC2008703 | Ning Xie |
| Shanghai Rising-Star Program | 22QA1408200 | Yanjing Zhu |
| Fundamental Research Funds for the Central Universities | 22120220555 | Yanjing Zhu |
| Fundamental Research Funds for the Central Universities | 22120230138 | Rongrong Zhu |
| Programs in Emergency and Critical Care Medicine, Shanghai Municipal Health Commission | KPB1702 | Liming Cheng |
| National Key Specialty Program, National Health Commission of China | GJ2301 | Liming Cheng |

The funders had no role in study design, data collection and interpretation, or the decision to submit the work for publication.

## Author contributions

Qing Zhao, Conceptualization, Resources, Data curation, Software, Formal analysis, Validation, Investigation, Methodology, Writing – original draft, Writing – review and editing; Yanjing Zhu, Conceptualization, Resources, Data curation, Formal analysis, Supervision, Funding acquisition, Validation, Investigation, Methodology, Writing – original draft, Project administration; Yilong Ren, Conceptualization, Resources, Data curation, Formal analysis, Supervision, Investigation, Methodology, Writing – original draft, Project administration; Lijuan Zhao, Software, Formal analysis, Validation, Investigation, Methodology; Jingwei Zhao, Haofei Ni, Formal analysis, Validation, Investigation, Methodology; Shuai Yin, Resources, Validation, Investigation, Methodology; Rongrong Zhu, Liming Cheng, Conceptualization, Supervision, Funding acquisition, Project administration, Writing – review and editing; Ning Xie, Conceptualization, Resources, Supervision, Funding acquisition, Visualization, Project administration, Writing – review and editing

## Author ORCIDs

Qing Zhao (ID) https://orcid.org/0000-0003-3437-533X
Yanjing Zhu (ID) https://orcid.org/0000-0002-2564-1176
Ning Xie (ID) http://orcid.org/0000-0002-0470-9928

Reviewer #1 (Public Review): https://doi.org/10.7554/eLife.95672.3.sa1
Reviewer #2 (Public Review): https://doi.org/10.7554/eLife.95672.3.sa2
Author response https://doi.org/10.7554/eLife.95672.3.sa3

## Additional files

### Supplementary files

• Supplementary file 1. Primers for transgenic mice genotyping and RT-PCR. (**a**) Transgenic mice and genotyping primers. (**b**) Primers for type I interferon signal pathway genes. (**c**) Primers for pro- and anti-inflammatory microglial marker genes. (**d**) Primers for Type I interferon (IFN) signal-related genes.

• MDAR checklist

### Data availability

The sequence and sample data have been deposited in the NCBI database under Sequence Read Archive (SRA) with Bioproject identification number PRJNA847704 (SRR19611667 - SRR19611678).

The following datasets were generated:

| Author(s) | Year | Dataset title | Dataset URL | Database and Identifier |
|---|---|---|---|---|
| Zhao Q, Zhu Y, Ren Y, Zhao L, Zhao J, Yin S, Ni H, Zhu R, Cheng L, Xie N | 2022 | Astrocyte elimination in the lumbar enlargement attenuates neuropathic pain after spinal cord injury | https://www.ncbi.nlm.nih.gov/bioproject/?term=PRJNA847704 | NCBI BioProject, PRJNA847704 |
| Zhao Q, Zhu Y, Ren Y, Zhao L, Zhao J, Yin S, Ni H, Zhu R, Cheng L, Xie N | 2023 | RNA-Seq of *Mus musculus*: adult female spinal cord | https://www.ncbi.nlm.nih.gov/sra/SRR19611667 | NCBI Sequence Read Archive, SRR19611667 |
| Zhao Q, Zhu Y, Ren Y, Zhao L, Zhao J, Yin S, Ni H, Zhu R, Cheng L, Xie N | 2023 | RNA-Seq of *Mus musculus*: adult female spinal cord | https://www.ncbi.nlm.nih.gov/sra/?term=SRR19611678 | NCBI Sequence Read Archive, SRR19611678 |

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
