## [Editor Report · eLife Assessment]

This **important** study demonstrated that ablation of astrocytes in the lumbar spinal cord not only reduced neuropathic pain but also caused microglia activation. The findings presented add considerable value to the current understanding of the role of astrocyte elimination in neuropathic pain, offering **convincing** evidence that supports existing hypotheses and insights into the interactions between astrocytes and microglial cells, likely through IFN-mediated mechanisms

---

## [Referee Report · Reviewer #1 (Public Review)]

Summary:

In this study the authors demonstrated that ablation of astrocytes in lumbar spinal cord not only reduced neuropathic pain but also caused microglia activation. Furthermore, RNA sequencing and bioinformatics revealed an activation of STING/type I IFNs signal pathway in spinal cord microglia after astrocyte ablation.

Strengths:

The findings are novel and interesting and provide new insights into astrocyte-microglia interaction in neuropathic pain. This study may also offer a new therapeutic strategy for the treatment of debilitating neuropathic pain in patients with SCI.

Weaknesses:

The authors have provided a satisfactory explanation of the comments on sample size, statistics, and the sex of the animals. The statistic was reworked.

---

## [Referee Report · Reviewer #2 (Public Review)]

Summary:

In the manuscript, Zhao et al. have carried out a thorough examination of the effects of targeted ablation of resident astrocytes on behavior, cellular responses, and gene expression after spinal cord injury. Employing transgenic mice models alongside pharmacogenetic techniques, the authors have successfully achieved the selective removal of these resident astrocytes. This intervention led to a notable reduction in neuropathic pain and induced a shift in microglial cell reactivation states within the spinal cord, significantly altering transcriptome profiles predominantly associated with interferon (IFN) signaling pathways.

Strengths:

The findings presented add considerable value to the current understanding of the role of astrocyte elimination in neuropathic pain, offering convincing evidence that supports existing hypotheses and valuable insights into the interactions between astrocytes and microglial cells, likely through IFN-mediated mechanisms. This contribution is highly relevant and suggests that further exploration in this direction could yield meaningful results.

Weaknesses:

The authors have satisfactorily addressed the comments regarding further clarifications and statistical methods.

---

## [Author Response]

The following is the authors’ response to the original reviews.

**Recommendations for the authors:**

**Reviewer #1 (Recommendations For The Authors):**
The manuscript could be improved by addressing the following issues.(1) Fig. 3: The analgesic effects after astrocyte ablation appear to recover after one week. Is this due to repopulation of astrocytes?

Although we did not detect the proliferation of astrocytes, we hypothesized that it was likely related to the microglia phagocytosis of astrocyte debris after astrocyte ablation. Microglia are known to have the function of phagocytosis of cell debris. Diphtheria toxin-mediated cell ablation caused AAV2/5-GfaABC1D-Cre labeled astrocytes death and cell fragmentation. We hypothesized that the microglia could phagocyte the astrocyte fragments and were stimulated to activate type I interferon signal. When microglia phagocyte debris ended, the activation of type I interferon signal was also declined. Reduced activation of type I interferon signal may also be accompanied by recurrence of pain.

(2) Fig. 3: Please justify the large sample size of n=30-36. Is this sample size based on previous studies or statistical estimation?

The number of mice was based on our previous report [1], and the increased number of mice may also ensure that the pain data would also be reliable. Not only did we explore the differences between the sexes, and we also needed to obtain samples at different times for different experiments.

(3) Please try to plot individual data points for some critical time points to demonstrate data distribution. It is also helpful to plot male and female data points separately for some time points.

Individual data have been plotted as your request and added in the supplementary material.

(4) It is unclear if the same number of males and females were used in this study, as females were typically used for SCI studies. I wonder if you can use repeated measures Two-Way ANOVA for statistical analysis.

According to our observations, the number of males and females was not the same, while both of them were sufficient for statistical analysis. In addition, in the process of breeding transgenic mice, we would obtain both male and female mice, and rational use of mice may be better for us. Indeed, previous studies have shown that female mice are more commonly used in pain studies. Although we did not observe a gender difference in this study, it has been reported in the previous studies that gender is one of the factors for pain differences. According to your suggestion, we adopted the Two-Way ANOVA for statistical analysis and updated it in the part of statistical methods, but the statistical results were consistent with the previous results, so we did not modify the statistical results of the pictures.

(5) Fig. 3C, D: The effects of astrocyte ablation on mechanical pain are mild, compared to thermal pain. Electronic von Frey apparatus may be difficult for mice. It works very well for rats and large animals.

Since the animals involved in this study were all mice, we did not know how electronic von Frey was used in rats and large animals. But after the using of electronic von Frey, it seems to us that electronic von Frey is very suitable for mouse experiments. Best of all, our electronic von Frey can achieve accuracy as low as 0.01g. This allows us to detect very sensitive pain data, which may be more accurate and intuitive than before.

(6) Fig. 2B: In the figure legend it states n = 3 biological repeats. There are many more dots in each column. Are these individual animals or spinal cord sections?

As we describe in our method, n = 3 biological repeats represented three biological repeats per group, i.e., three mice/group with three IF per mouse. We take three or more values in each ascending tract (depending on the partition size of the different ascending tracts of lumbar enlargements). So, we would get more data as shown in Figure 2, which could be also more reliable.

(7) Fig. 4C: It appears that GFAP is increased by toxin treatment. Please explain this result.

This figure was calculated for astrocyte activation in the lesion area (T9-10), but not for the lumbar enlargement.

**Reviewer #2 (Recommendations For The Authors):**
Specific Comments:RNA-Sequencing Analysis: The strength of the RNA-sequencing data in elucidating the impact of astrocyte elimination is compelling. While the focus on IFN signaling is well-supported, the manuscript overlooks other differentially expressed genes. A deeper analysis or at least a discussion of these genes could enrich the study's conclusions, offering a more holistic view of the underlying mechanisms.

Although we did not focus more on other relevant differential genes, we focused on the most significant differential genes, for these differential genes have a more significant effect on pain.

Q2: Figure Presentation: Consolidating Figures 1-3 could increase the clarity of the result presentation, reducing distractions from the main narrative. Certain aspects, such as the comparison of different tracts in Figure 2B and the body weight data in Figure 3C, seem tangential and might be better suited for supplementary materials.

The comparison of astrocyte activation in different ascending tracts of lumbar enlargements explained the relationships between astrocyte activation and pain, and laid the foundation for the subsequent astrocyte elimination. The weight data is also important, reflecting not only the changes in the overall recovery process after spinal cord injury, but also the effect of astrocyte elimination on the overall effect of mice. Thus, the weight data together with the pain test results will be more intuitive for the reader to understand the change of overall conditions of mice after astrocyte elimination.

Q3: Schematic Clarity: The schematic in Figure 1A is confusing, particularly in distinguishing between transgenic mice and viral constructs. The inconsistent naming of Cre recombinase (alternatively referred to as Cre, CRE, and sometimes DRE) further complicates understanding. Standardizing these elements would greatly enhance clarity for the readers.

As we described in the part of method, *Gt(ROSA)26Sorem1(CAG-LSL-RSR-tdTomato-2A-DTR)Smoc* mice contain both Loxp-stop-Loxp sequence and Rox-stop-Rox sequence. In the process of reproduction, *Gt(ROSA)26Sorem1(CAG-LSL-RSR-tdTomato-2A-DTR)Smoc* mice crossed with C57BL/6JSmoc-Tg(CAG-Dre)Smoc mice could remove the Rox-stop-Rox sequence, which could further crossed with mice containing Cre recombinase, or with AAV2/5-GfaABC1D-Cre intervention to remove the Loxp-stop-Loxp sequence and induce the expression of tdTomato and DTR.

Q4: Pathway Analysis: The discussion of the signal pathway analysis in Figure 8 leans heavily on speculation without direct evidence from the study. Distinguishing clearly between findings and literature-derived hypotheses is crucial. A more detailed discussion that properly cites sources for each pathway element would strengthen the manuscript.

According to your question, we have added this figure to the supplementary picture.

Q5: Statistical Analysis: The use of one-way ANOVA, despite presenting data in groups, is misaligned with the data's structure. Employing two-way ANOVA followed by post-hoc comparisons is appropriate for statistical analysis.

According to your suggestions, we adopted the Two-Way ANOVA for statistical analysis and updated it in the part of statistical methods, but the statistical results are consistent with the previous ones. Therefore, we did not modify the statistical results of the pictures.